# Voltage Sag Causes Recognition with Fusion of Sparse Auto-Encoder and Attention Unet

**Rui Fan** [1], **Huipeng Li** [1], **Tao Zhang** [2], **Hong Wang** [2,*], **Linhai Qi** [2] and **Lina Sun** [2]

1 State Grid Electric Power Research Institute of Shanxi Electric Power Company, Taiyuan 030001, China
2 School of Control and Computer Engineering, North China Electric Power University, Changping District, Beijing 102206, China
* Correspondence: wh@ncepu.edu.cn

**Abstract:** High-precision voltage sag cause identification is significant in solving the power quality problem. It is challenging for traditional deep learning models to balance training complexity and recognition performance when processing high-dimensional staging data samples, which affects the final recognition effect. This paper proposes a voltage sag identification method that fuses a sparse auto-encoder and Attention Unet. The model uses a sparse auto-encoder to perform unsupervised feature learning on the high-dimensional voltage sag waveform data and automatically obtains the deep low-dimensional features. Attention Unet, fused with cross-layer spatial and channel attention modules, further extracts these features to obtain recognition results with high performance. Compared with other deep learning recognition methods, the noise-adding experiments and the measured data are verified, indicating that the proposed method has low training complexity, higher recall, and better noise immunity. It benefits auxiliary decision-making for power quality management and governance.

**Keywords:** voltage sag; sparse auto-encoder; Unet; attention mechanism

## 1. Introduction

With the rapid development of smart grids, the increasing complexity of power loads, and the large-scale integration of nonlinear disturbance sources into the power grid, more and more sensitive equipment is being used by end users. The power quality problems have become prominent, and a large number of power quality disturbance events have caused a severe impact on industrial production and the lives of users [1–5]. Among them, the economic loss caused by voltage sag to sensitive users is also even more significant, and it has become a critical power quality problem [6]. In addition, with the increasing complexity of power load and the continuous improvement of precision instruments for the stability of the power system, the voltage sag problem has gradually attracted the attention of academia and industry [7]. High-precision voltage sag classification can provide auxiliary decision-making for power quality management and governance [8], which is significant for solving power quality problems.

There are two main voltage sag causes recognition methods according to the feature extraction process of the algorithm. The first is to extract the features of the perturbation signal through manual mechanism analysis and then input them to the corresponding machine learning classifier for sag cause identification. In the face of an increasingly complex power environment, it is not easy to make a comprehensive and accurate analysis of the staging signal based on mechanism analysis. The second is the models automatically extract abstract features, usually based on the end-to-end deep learning. The purpose of comprehensively extracting features and accurately identifying voltage sags in deep networks is achieved through the characterization learning of multi-level networks. For the first method, feature extraction algorithms include S transform [9], wavelet transform [10], fast

Fourier transform [11], and Hilbert–Huang transform (HHT) [12]. Machine learning classification algorithms include clustering algorithm, support vector machine [13], principal component analysis [14], decision tree (DT), etc. Reference [9] proposed a voltage sag classification method based on the similarity of standard templates based on S transform. It can classify single fault self-clearing, single-phase short circuit, fault type change, transformer excitation, and asynchronous motor starting. Reference [15] proposes a power quality disturbance recognition method based on HHT and mainly identifies three types of sags: line short circuit fault, transformer energization, and large capacity induction motor startup. Compared with other mechanisms, the model has a specific recognition performance. Still, the model relies on the disturbance signal's characteristics and typical working conditions. Making a unified judgment on the increasingly complex power environment is difficult, and the generalization ability is weak [16]. For the second method, the data-driven deep neural network does not rely on the characteristics of the traditional physical model. It can automatically learn from massive feature data and analyze complex internal patterns [17], which can also effectively make up for the lack of adaptability of the method based on the mathematical statistical model. In the reference [18], a recognition model of a deep belief network (DBN) is proposed to achieve intelligent classification recognition of voltage sag. The deep learning recognition model has achieved high recognition performance and good generalization ability. Reference [19] uses a CNN to extract the temporal and spatial characteristics of staging signals. Those data-driven methods can complete automated feature learning and hierarchical feature extraction to overcome the problem of manual acquisition of features [20]. However, it is difficult to deal with high-dimensional staging datasets. When the number of original staging samples and the dimension of the feature vector are large, too many training parameters and higher algorithm complexity will affect the recognition ability.

To reduce the complexity of high-dimensional feature training and improve the performance, reference [21] maps high-dimensional raw data to low-dimensional hidden variable vector space by training auto-encoders (AEs) and finally outputs the perturbation type through a single-layer convolutional neural network. Reference [22] realizes dimension reduction learning of high-dimensional features by stacking sparse auto-encoder (SAE), and can accurately identify nine single disturbances with noise and four composite power quality disturbances. However, ordinary convolutional networks are used for feature extraction to lose the essential original information during training and thus affect the classification effect. Attention Unet (AttUnet) adopts a symmetrical encoded and decoded network structure in the design of multiple convolutional neural networks, which has an excellent performance in the fields of fine feature segmentation [23]. The AttUnet encoder can extract the underlying feature data at a deep depth and the decoder can extend up to obtain the high-level feature data. The low-level features and the high-level features are fused through a layer-by-layer jump connection [24]. Before the jump connection is performed, the attention-gating mechanism is combined to control the importance of the features of different dimensions to improve the sensitivity of the neural network. Without significantly increasing the training parameters, the network improves the performance and learning ability of feature extraction and makes the features obtained by the model more expressive, which provides a reliable idea for the final classification.

Based on above deep learning models, this paper proposes a voltage sag recognition method that fuses SAE and AttUnet. First, the SAE is used to automatically extract features from unlabeled voltage sag data, learn the prosperous staging state in the high-dimensional staging dataset, and condense the raw data of high-dimensional into low-dimensional high-precision feature expression. Then, AttUnet further extracts the in-depth features for low-dimensional staging data. The feature maps of the encoder and decoder output are combined and superimposed by combining the layer-by-layer jump connection and the spatial and channel attention mechanism to ensure the performance and learning ability of the feature extraction model. The extracted feature data are more expressive and more distinguishable, to be easy to identify.

## 2. Voltage Sag Characteristic Analysis

Voltage sag refers to the instantaneous drop of the root mean square (RMS) value of voltage to 90~10% of rated voltage amplitude and recovery after 0.5~30 cycles [25]. Natural disasters or load transient conditions may result in voltage sags [26]. Climate, equipment failure, birds nesting across the distribution lines, and line short circuit failures will lead to different short-circuit faults. However, identifying whether the short-circuit fault is caused by weather, equipment failure, a bird's nest, or others is still challenging. Therefore, all literature only analyzes short-circuit fault, subdivided into single-phase, bi-phase, and three-phase. The sag of a large motor start or the switching of the transformer has its characteristics and can identify the corresponding type. Since the short-circuit fault, large motor start-up, and transformer switching in the power system are the leading causes of the voltage sag [27], this paper mainly studies these three voltage sags.

A complete staging event is a continuous waveform with time persistence. Different types of staging have other characteristics and tend to change significantly during the descent and recovery of perturbations [28]. As shown in Figure 1, in the system grounding fault, the RMS signal of the voltage sag shows a step-like drop and recovery more rapidly. During the voltage sag, the sag amplitude of the three-phase voltage is directly related to the phase where the fault occurs. For the induction motor start, its voltage sag RMS signal drops fast, and after dropping to the lowest point, it starts to recover slowly and shows frequent small fluctuations. Meanwhile, the amplitude of the three-phase voltage changes synchronously. For fault of the transformer switching, the three-phase RMS signal drops vertically and lasts for a while at the lowest point. After that, it recovers quickly to a certain amplitude, then slowly, so the whole recovery period is long. Furthermore, the three-phase voltage sag has different amplitude after the transformer switching, and the signal contains harmonic components, resulting in sharp bumps in the RMS.

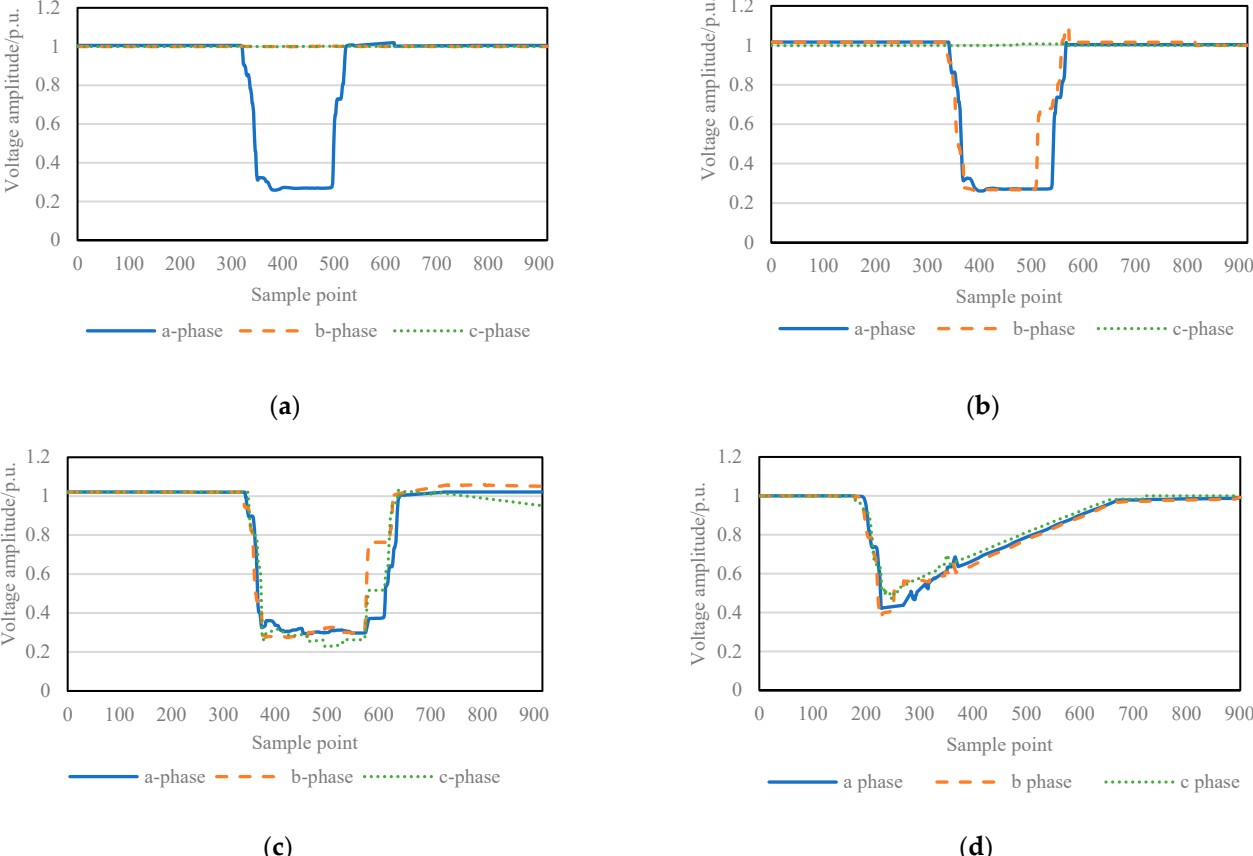

**Figure 1.** *Cont.*

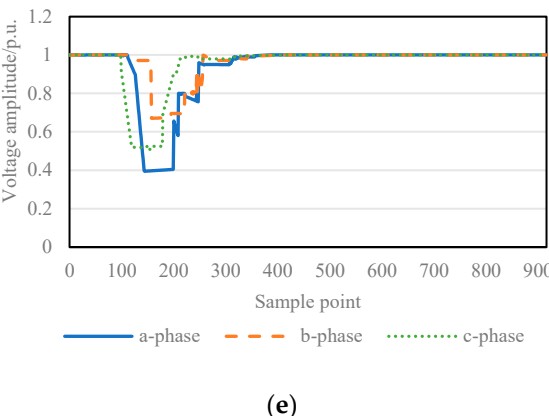

(**e**)

**Figure 1.** Comparison chart of different types of sag signals. (**a**) Voltage sag is caused by a single-phase grounding fault; (**b**) voltage sag is caused by a two-phase grounding fault; (**c**) voltage sag is caused by a three-phase grounding fault; (**d**) voltage sag is caused by induction motor starting; (**e**) voltage sag is caused by transformer switching.

In summary, each type of temporary drop in RMS signal has distinct variation characteristics. This paper calls the original provisional drop recording data with large dimensions a high-dimensional sample, affected by noise factors such as power grid conditions, and the same kind of features have instability in the original feature space. In contrast, the larger dimension leads to increased training complexity, which is not conducive to efficient identification. Deep learning can extract more effective depth characteristics of high-dimensional samples to give the model better generalization ability.

## 3. Implementation of the Model

The structure diagram of the integrating SAE and AttUnet is shown in Figure 2. SAE is a symmetric, unsupervised 3-layer neural network that contains encoders and decoders. The encoder can map high-dimensional feature spaces to low-dimensional implicit layer vector spaces to reduce feature dimensionality. The decoder reverses the implicit layer vector space to the output layer features and implements feature reconstruction to fit the nonlinear coupling relationship between the input layer data of the sample staging signal and the reconstructed output layer data. SAE adds sparse constraint terms to the loss function, improves feature learning ability [29], and obtains a more efficient low-dimensional representation of high-dimensional staging features in the implicit layer. The low-dimensional data enter AttUnet for further feature extraction. AttUnet is a symmetrical U-shaped full-scale convolutional neural network that introduces a contextual attention mechanism. Consisting of symmetrical encoders and decoders and jump connections connecting low-level features with higher-level features, it is possible to learn elements of different scales and semantic levels. Controlling the importance of features in different dimensions through the attention learning mechanism can effectively reduce the influence of noise on parts and obtain features with higher differentiation and more expressiveness. Finally, they enter the multi-layer fully connected layer and the Softmax layer for classification. In this paper, the fusion model achieves efficient dimensionality reduction on high-dimensional sample drop data through SAE, which can significantly reduce the training complexity and improve the training accuracy. At the same time, the model characteristics of AttUnet are used to synthesize the high-level and low-level feature data in extracting features, retain critical information, improve feature differentiation, and improve the fusion model's recognition effect.

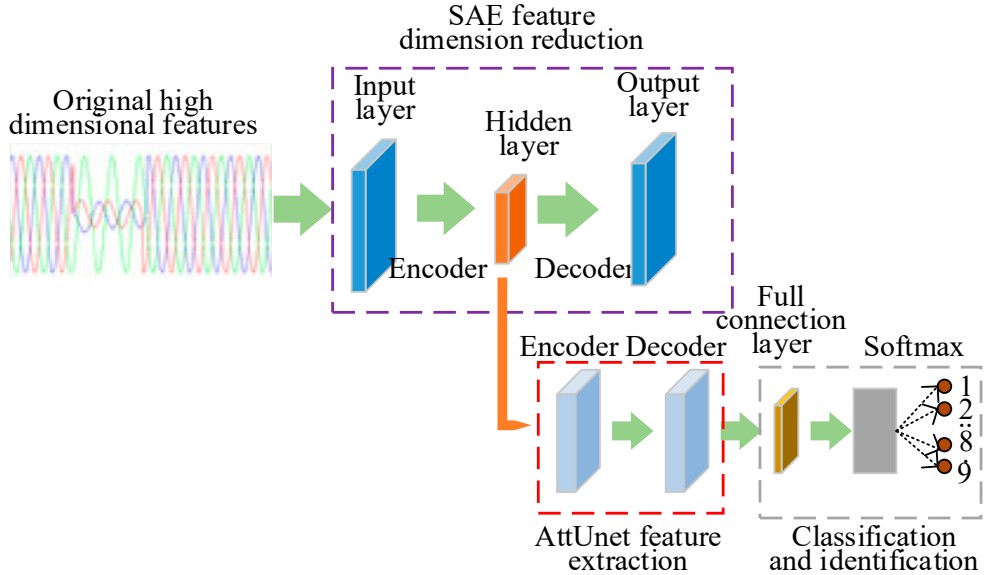

**Figure 2.** Structure diagram of voltage sag recognition model integrating SAE and AttUnet.

### 3.1. Feature Dimension Reduction Based on SAE

High-dimensional staging data contain abundant staging state information, which plays an essential role in accurate identification. When traditional pattern recognition methods process high-dimensional data, they are prone to dimension disaster, and model training is too complicated. SAE compresses and maps the original high-dimensional data through an encoder to obtain a low-dimensional hidden variable space and realize data dimensionality reduction. Then, the decoder is used to restore the implicit layer vector space to the output feature to realize data reconstruction. SAE can automatically learn the prosperous staging state in the high-dimensional staging data samples and form a sequence-to-sequence mapping nonlinear coupling relationship with the encoder and decoder. The comprehensive encoder and decoder form a sequence-to-sequence mapping nonlinear coupling relationship. As shown in Figure 3, AE is mainly divided into encoders and decoders. The encoder maps the input data to the low-dimensional hidden variable space through multiple convolutional layers. The decoder restores the input data by reverse mapping the hidden variable space to the output layer through the multi-layer deconvolution layer and convolution layer. The training process is to fit the nonlinear coupling relationship between the input layer data and the reconstructed output layer data, and finally condense it into the low-dimensional feature "essence". The output of the hidden layer space feature vector is used as the result of the model for classification and recognition, which can significantly reduce the training complexity and improve the training accuracy.

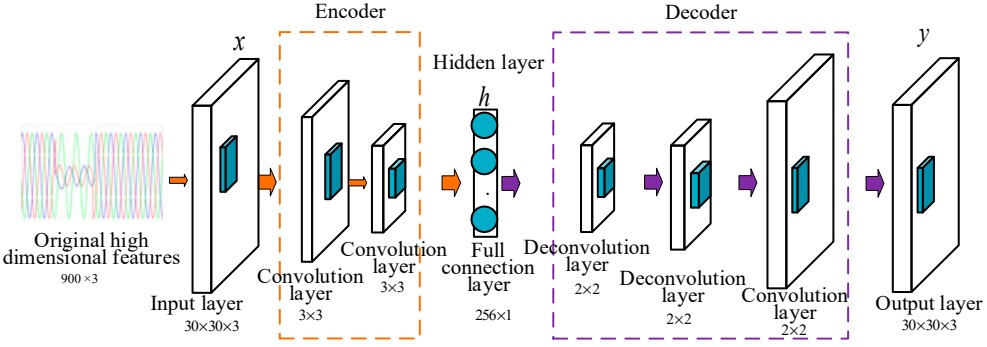

**Figure 3.** AE network structure diagram.

Assuming that the unsupervised training set is $X(x_1, x_2, \ldots, x_n)$, $n$ is the number of samples, and $g$ is the encoder function, the encoder is calculated as follows:

$$h = g(W_1 x + b_1), \tag{1}$$

where $h$ is the vector feature of the implicit layer in the middle, $W_1$ is the weight vector of the encoder, $x$ is the original feature of the input sample, and $b_1$ is the bias vector of the encoder.

$f$ as a decoder function, the decoder is calculated as:

$$y = f(W_2 h + b_2), \tag{2}$$

where $W_2$ and $b_2$ are the weight vector and the bias vector of the decoder, $y$ is the output characteristic of the decoder refactoring. The characteristic dimensions of $x$ and $y$ are the same, and the training goal of AE is to achieve that $x$ and $y$ not only have the same size but also minimize the data differences at the corresponding feature points.

The encoder contains two convolutional layers. They have the same size kernel of $3 \times 3$, but different steps 1 and 2 in order. The fully connected layer of 256 neurons is connected after the convolutional layer as the implicit layer. The decoder consists of two deconvolutional layers and one convolutional layer. The kernel size of the deconvolutional layer is $2 \times 2$, and the steps are 1 and 2, respectively. The kernel size of the convolutional layer is $2 \times 2$, and the step size is 1.

AE, through cyclic iteration, updates the weights and biases of the neural network to achieve the encoder and decoder sequence-to-sequence coupling relationship training. Therefore, the output voltage sag data can stay consistent with the input voltage sag data. In this paper, to minimize the mean absolute error (MAE) loss function as the goal, the loss function is:

$$M(W, b) = \frac{\sum\limits_{i=1}^{n} \|x_i - y_i\|}{n}, \tag{3}$$

where $x_i$ is the $m$-dimensional input features of the $i$-th samples, and $y_i$ is the $m$-dimensional output features of the reconstructed $i$-th samples. SAE is an extension of AE. To make AE have better performance in extracting features, sparse constraint units are added based on the hidden layer, which reduces the activity of neurons. Only some neurons are activated and learned, which further improves its characterization learning ability. Thus, the loss function of the SAE is:

$$M_*(W, b) = M(W, b) + \alpha \frac{\sum\limits_{q=1}^{s} \left[ \rho \ln \frac{\rho}{p^q} - (1 - \rho) \ln \frac{1-\rho}{1-p^q} \right]}{s}, \tag{4}$$

where $\alpha$ is the weight of the sparsity penalty factor; $S$ is the number of neurons in the implicit layer; $\rho$ is a sparseness parameter, usually set very small close to 0; $p^q$ is the average activation of the $q$-th neuron in the implicit layer on the training set. When the loss error of the model, $M_*(W, b)$, is the smallest, a pre-trained SAE-based voltage sag characteristic dimensionality reduction model is obtained. The encoder accepts the original high-dimensional data to receive the characteristics of the hidden layer of $256 \times 1$ and then reconstructs it into a matrix of $16 \times 16 \times 1$ by reshaping and training on behalf of the convolutional network AttUnet.

*3.2. Sag Recognition Based on AttUnet*

Based on AttUnet, this paper extracts and identifies the staging features. Through its unique encoding–decoding method and jump transfer structure, it can effectively extract detailed features at full scale. Combined with the attention-gating mechanism, we learn the features of different scales and different semantic levels, extract the expressive and differential features, and, finally, input to the multi-layer fully connected layer for classification.

Figure 4 shows the model structure. The encoder contains two convolutional blocks and two down sampling layers, each convolutional block consisting of two convolutional layers with a convolutional kernel size of $3 \times 3$. The top pooling layer achieves down sampling with a convolutional core of $2 \times 2$. By extracting downwards, the network obtains underlying features. The decoder contains three convolutional layers and two up sampling layers. The web can scale up to obtain high-level feature data, and when decoding operations, it can recover the feature size by using $2 \times 2$ up sampling. Once the up sampling process is completed, the obtained features are fused with the feature map extracted by the previous layer. Performing the $3 \times 3$ convolution operation again restores the feature details, making the edge features more refined and improving the sensitivity. This model has two cross-layer hop connections to fuse the underlying and high-level features and introduces a contextual attention guidance mechanism (AG) before each jump connection. By controlling the importance of different dimensional characteristics, the sensitivity of neural networks is improved. The performance and learning ability of network feature extraction are improved without significantly increasing the training parameters. The convolutional layers of the web all use the ReLU activation function and batch normalization to accelerate learning. The fully connected layer contains two hidden layers with 16 and 9 neurons, respectively. The activation function uses tanh and finally maps the output type probability to the [0, 1] interval via the Softmax layer, thus classifying the voltage sag.

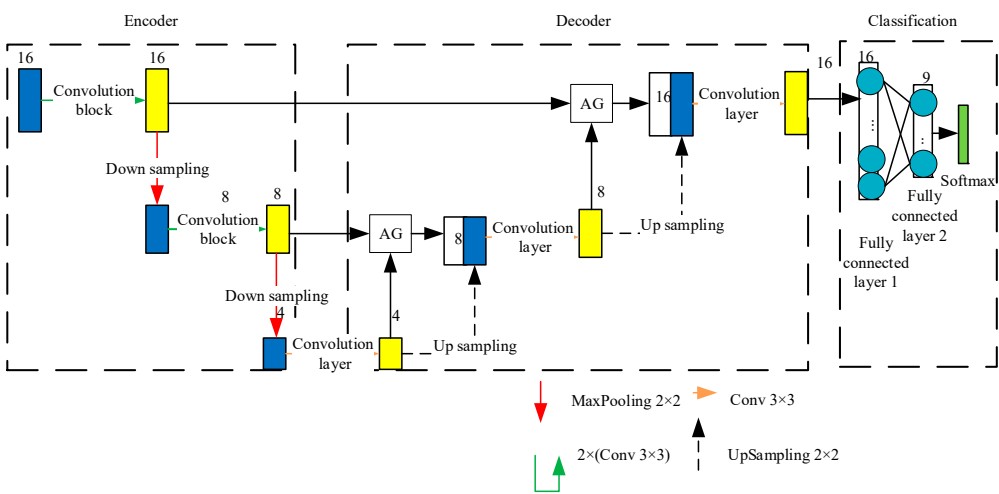

**Figure 4.** Structure diagram of sag recognition model based on AttUnet.

AG is a spatial attention module, its structure as shown in Figure 5. *g* is the decoder computed high-level features, and *x* is the encoder calculated underlying features. *g* and *x* are respectively added after convolution to obtain the attention feature, and then successively pass through the ReLU activation function, a $1 \times 1 \times 1$ convolution layer, and sigmoid activation function to finally obtain the soft attention feature *attr*. To have the same feature size as *x*, *attr* needs to be resampled. Multiplying the updated *attr* by the *x* of the jump connection part can enhance the sag feature spatially and control the importance of features in different positions, which can powerfully lessen the impact of noise on feature extraction.

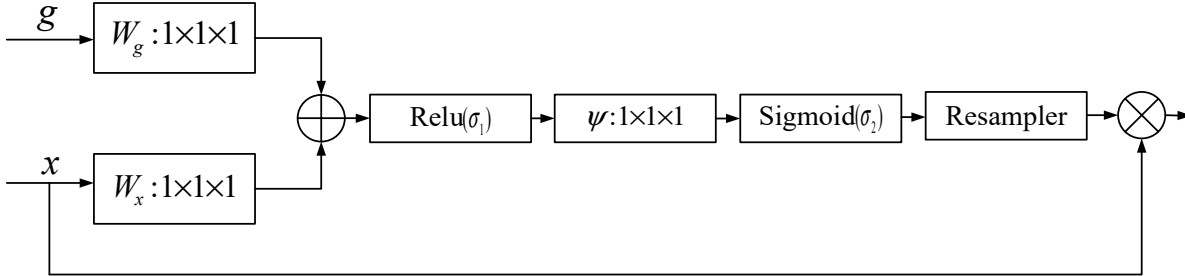

**Figure 5.** AG module structure diagram.

Its mathematical expression is:

$$attr = \sigma_2(\psi(\sigma_1(W_x x + W_g g + b_g)) + b_\psi),\tag{5}$$

where $\sigma_1$ is the ReLU function, $\sigma_2$ is the sigmoid function, $W_x$, $W_g$, $\Psi$ are convolutional layers, $b_g$, $b_\Psi$ are bias terms, and through the above formula can obtain a soft attention feature *attr* compressed between [0, 1].

This section maps the decoder output of AttUnet through the 2-layer fully connected layer and the Softmax layer into the [0, 1] interval to obtain the probabilities of each type to perform multi-label classification. Assuming that there are $K$ sag types, $S_c$ represents the probability that the recognition type is $c$, $e^{a_k}$ represents the output of the $k$-th neural unit, and the Softmax layer is calculated as follows:

$$S_c = \frac{e^{a_c}}{\sum\limits_{k=1}^{K} e^{a_k}}.\tag{6}$$

The loss function $L$ of the classification model adopts the cross-entropy loss function, the mathematical formula of which is:

$$L = -\sum_{i=1}^{n} y_i \log(\hat{y}_i),\tag{7}$$

where $y_i$ represents the accurate classification result, $\hat{y}_i$ represents the predicted classification result, and $n$ is the number of samples. When the loss error $L$ of the network structure is the smallest, a pre-trained AttUnet-based voltage sag recognition model is obtained.

In summary, this paper shows high-precision classification and identification of voltage sag by fusing SAE and AttUnet. SAE is used to learn the prosperous staging operation state in the staging characteristics of high-dimensional samples, fit the nonlinear relationship between input and reconstructed output, and condense it into low-dimensional feature "essence" for classification and identification. The "essence" of the features is further extracted by AttUnet, and the lower-level features and the higher-level features are integrated to ensure that the key information is retained in the training process, improving the discrimination between the sag features. Then, substituting them into the classifier for identification can reduce the training complexity to a certain extent and improve the recognition accuracy and enhance anti-noise performance of the model.

## 4. Study Analysis

Since the staging recognition model requires a large amount of labeled voltage sag data for analysis training, this study obtains experimental data by simulating experiments.

### 4.1. Experimental Dataset and Evaluation Index

There are two experimental datasets. One is the simulated data and the other is the measured data.

The simulated experiment builds the IEEE14 node system through simulation software. The fundamental frequency is set to 50 Hz, the system is composed of power components such as a power supply, monitoring node, power line, and transformer. All monitoring points 1 to 14 perform sag wave recording, and monitoring nodes 1, 2, 3, and 8 are all connected to the power supply. The system structure diagram is shown in Figure 6.

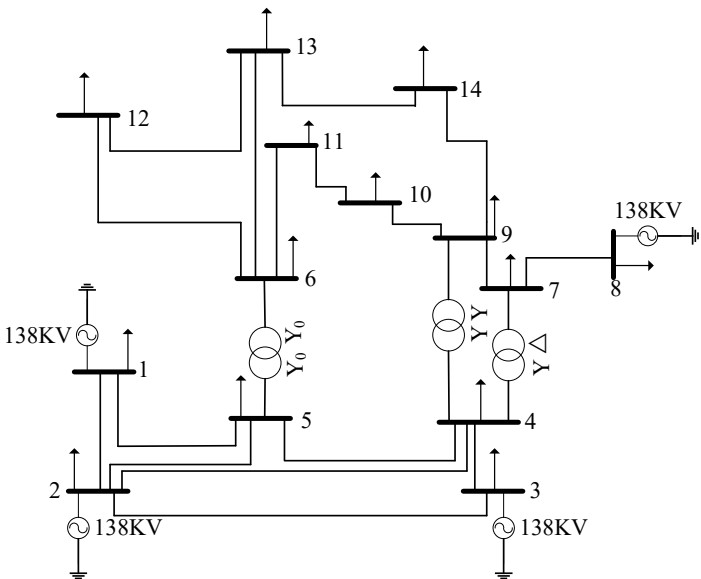

**Figure 6.** IEEE14 node system.

By adjusting the line impedance of the node system, the type of fault sag, the time of fault start and end, the location of the fault, the transformer, and other simulation parameters, and setting the recording duration at 0.3 s, the software obtains wave recording data in all monitoring points. The a, b, and c three-phase voltages are recorded at the same time. There are a total of 7 short-circuit faults (a phase, b phase, c phase, ab phase, ac phase, bc phase, abc phase, recorded as C1-C7), large motor start (recorded as C8), and transformer switching (registered as C9). There are 500 recorded waveform data for each sag cause. The characteristic data length is 900 sample points and three phases, and the corresponding SAE model input data interface size is $30 \times 30 \times 3$.

There are a total of 4500 tentative datasets in this experiment. Seventy-five percent of the data in each staging signal feature are used for model training, and 25% for model verification. The simulation experiment adopts four-fold cross-verification and takes the average recall of the validation set as the final evaluation index. Recall, also called sensitivity or true positive rate (TPR), refers to the proportion of the positive samples (the current sag cause's samples) and the formula is:

$$\text{recall} = \text{TPR} = \frac{TP}{P} \times 100\%, \tag{8}$$

where $P$ is the actual sample size of one sag cause type, and $TP$ is the sample size that was correctly recognized.

Since the actual staging signal will be disturbed by noise, to illustrate the practical application effect of this model, this study adds Gaussian white noise with different signal-to-noise ratios, 50 dB, 40 dB, 30 dB, and 20 dB, to the staging signals of the original verification dataset to ensure the good anti-interference ability of the identification model. The noise formula for waveform sample $x_i$ is:

$$noise = Random(m) \cdot \sqrt{\frac{\sum\limits_{j=1}^{m} x_{ij}^2}{m \cdot 10^{\frac{SNR}{10}}}} \times 100\%, \tag{9}$$

where $m$ is the number of sample points, namely the $m$ dimension of waveform sample $x_i$, $x_{ij}$ is the $j$-th dimension of $x_i$, and *SNR* is the signal-to-noise ratio. Accordingly, when *SNR* becomes smaller, the noise becomes bigger. Table 1 shows the waveforms of C1, C4, C7, C8, and C9 under different noises.

**Table 1.** The waveforms of different voltage sag types under different noises.

| Sag Type | No Noise | *SNR* = 50 dB | *SNR* = 30 dB | *SNR* = 20 dB |
|---|---|---|---|---|
| C1 |  |  |  |  |
| C4 |  |  |  |  |
| C7 |  |  |  |  |
| C8 |  |  |  |  |
| C9 |  |  |  |  |

The measured dataset comes from the voltage sag waveform data of a power grid from January to September 2021. Due to the regional characteristics of the sag, only 53 short-circuit type voltage sags were captured during this period in the region, including four b-phase short-circuits (C2), ten bc-phase short-circuits (C6), and thirty-nine abc-phase short-circuits (C7).

The measured dataset is very imbalanced, and recall does not perform well with imbalanced datasets. Accordingly, we add the balanced accuracy of each sag cause as its index, the formula is:

$$\text{Balance Accuracy} = \frac{\text{TPR} + \text{TNR}}{2} \times 100\%, \tag{10}$$

where true positive rate (TPR) refers to the proportion of the positive samples (the current sag cause's samples) that are correctly identified. True negative rate (TNR) refers to the proportion of negative samples (all the other sag causes' samples) that are correctly identified.

### 4.2. Network Training and Process Analysis

This study first takes the original data 900 × 3-dimensional data to reshape them into 30 × 30 × 3-dimensional temporary drop data. They then enter the SAE for feature extraction, and the encoder calculates the original data and outputs them to the implicit layer to obtain a low-dimensional 256 × 1-dimensional data feature. After SAE training, the raw data can better remove redundant variables, extract key features, and the obtained features are easier to classify, which can prevent the overfitting of the recognition model and improve the generalization ability of the recognition model to a certain extent. Figure 7 shows the SAE model iteration process and, with the increase in the number of iterations, the loss rate of the model gradually decreases and becomes more stable. The model maintains a low error loss rate and a high degree of model fit, which reflects the excellent dimensionality reduction ability of the SAE model.

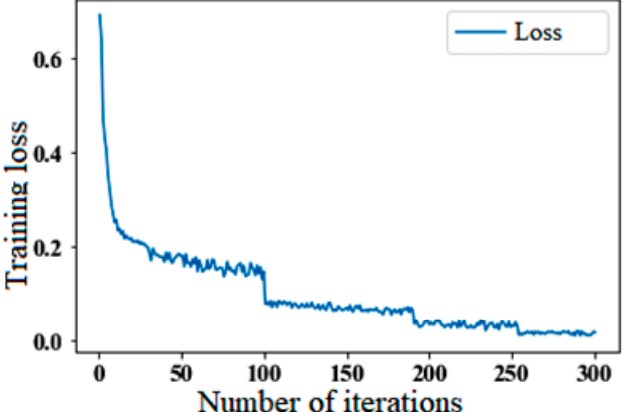

**Figure 7.** SAE model iteration process.

The experiment then fixed the parameters of the SAE model, used the implicit layer output data of the training set after encoder feature extraction to reconstruct a 16 × 16 × 1 three-dimensional matrix, and then entered it into the fully convolutional neural network AttUnet for classification recognition training. In Figure 8, as the number of training iterations increases, the accuracy increases at a gentle rate and tends to 100%, and the loss rate decreases and tends to be stable. SAE compresses high-dimensional features, extracts low-dimensional data expression, and then trains through AttUnet, which can significantly reduce the complexity of classification model training, prevent model overfitting, reduce iteration times, and improve accuracy.

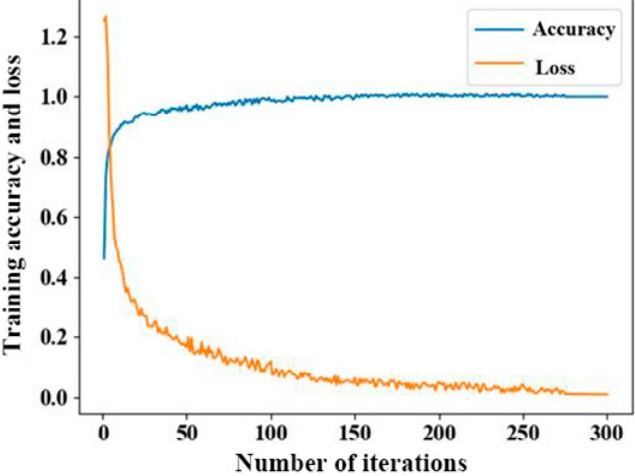

**Figure 8.** Iterative process of classification model.

### 4.3. Model Validation

To verify the model's performance under different noise conditions and natural environments, the Gaussian white noise of 50 dB, 40 dB, 30 dB, and 20 dB was added to the simulation sample data. Table 2 shows the experimental results and takes the average recall of four-fold cross-verification. With the superposition of noise, the input data are close to the actual data, and the model in this paper can effectively process the data. The experiment maintains a good classification effect and has good generalization ability, which verifies the effectiveness and reliability of the model.

**Table 2.** Classification results of the model under different noises.

| Voltage Sag Type | Average Recall/% | | | | |
|---|---|---|---|---|---|
| | No Noise | SNR = 50 dB | SNR = 40 dB | SNR = 30 dB | SNR = 20 dB |
| C1 | 100.0 | 100.0 | 100.0 | 99.56 | 99.31 |
| C2 | 100.0 | 100.0 | 100.0 | 100.0 | 98.89 |
| C3 | 100.0 | 99.70 | 99.00 | 98.88 | 98.55 |
| C4 | 98.85 | 100.0 | 100.0 | 98.50 | 98.21 |
| C5 | 100.0 | 100.0 | 99.80 | 99.80 | 99.40 |
| C6 | 100.0 | 99.50 | 98.73 | 98.22 | 97.31 |
| C7 | 98.97 | 99.88 | 99.51 | 98.50 | 98.33 |
| C8 | 99.82 | 99.70 | 99.00 | 98.88 | 98.84 |
| C9 | 100.0 | 99.53 | 99.40 | 98.50 | 98.00 |

The measured samples can be directly verified by bringing in the trained model, which can reflect the excellent application ability of the model. In the measured data, the recall of C2 and C6 types is 100%, and the recall of C7 types is 97.44%. Among them, the model identifies one C7 type error as C2. The confusion matrix shows the details in Figure 9 and the balance accuracy of C2, C6, and C7 is 98.98%, 100%, and 98.72%. Through verification and analysis, the proposed model has a high recognition accuracy for the measured data.

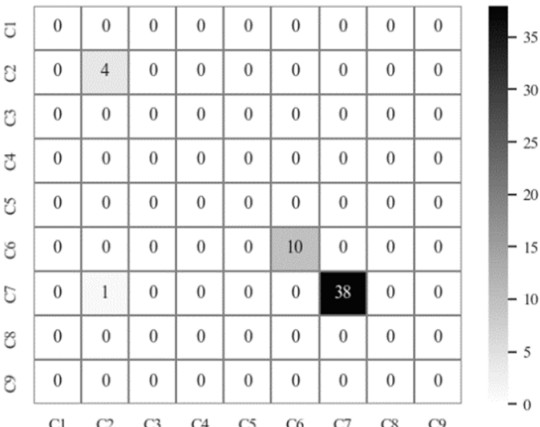

**Figure 9.** Confusion matrix for measured data recognition.

### 4.4. Comparison with Other Deep Learning Models

Recently, some deep learning models have been applied to identify voltage sags, and corresponding model designs are proposed based on different staging signal characteristics, which have good recognition capabilities. Considering the complex working conditions of the actual operation of the power grid, the short-circuit fault data were measured under the *SNR* of 50 dB, 40 dB, 30 dB, and 20 dB, respectively. The larger the *SNR*, the higher the recognition accuracy of the model, and vice versa. When the noise increases,

it has an effect on any model, and the accuracy decreases. Compared with the existing classification recognition algorithms of AttUnet, LSTM + Attention [27], LSTM [30], and SAE + CNN [21], the experimental results illustrate the applicable practical effects of the proposed model. As shown in Figure 10, although the *SNR* decreases to 20 dB, the average recall of SAE + AttUnet is more than 97.5%. That means the model is relatively stable under different working conditions. Its average recall remains in the range of 97.5–100% under various noise conditions, the individual AttUnet model is 96–99%, and the SAE + CNN model is in the range of 94–98.2%. The widely used LSTM model is in the range of 94.3–97.6%. After adding the attention mechanism, LSTM + Attention is greatly improved, and its recognition accuracy is 96–100%. Under the same number of iterations, based on the average accuracy, the ranking of the models is as follows: SAE + AttUnet, LSTM + Attention, AttUnet, LSTM, SAE + CNN. Therefore, the average accuracy of this model is higher than that of other models.

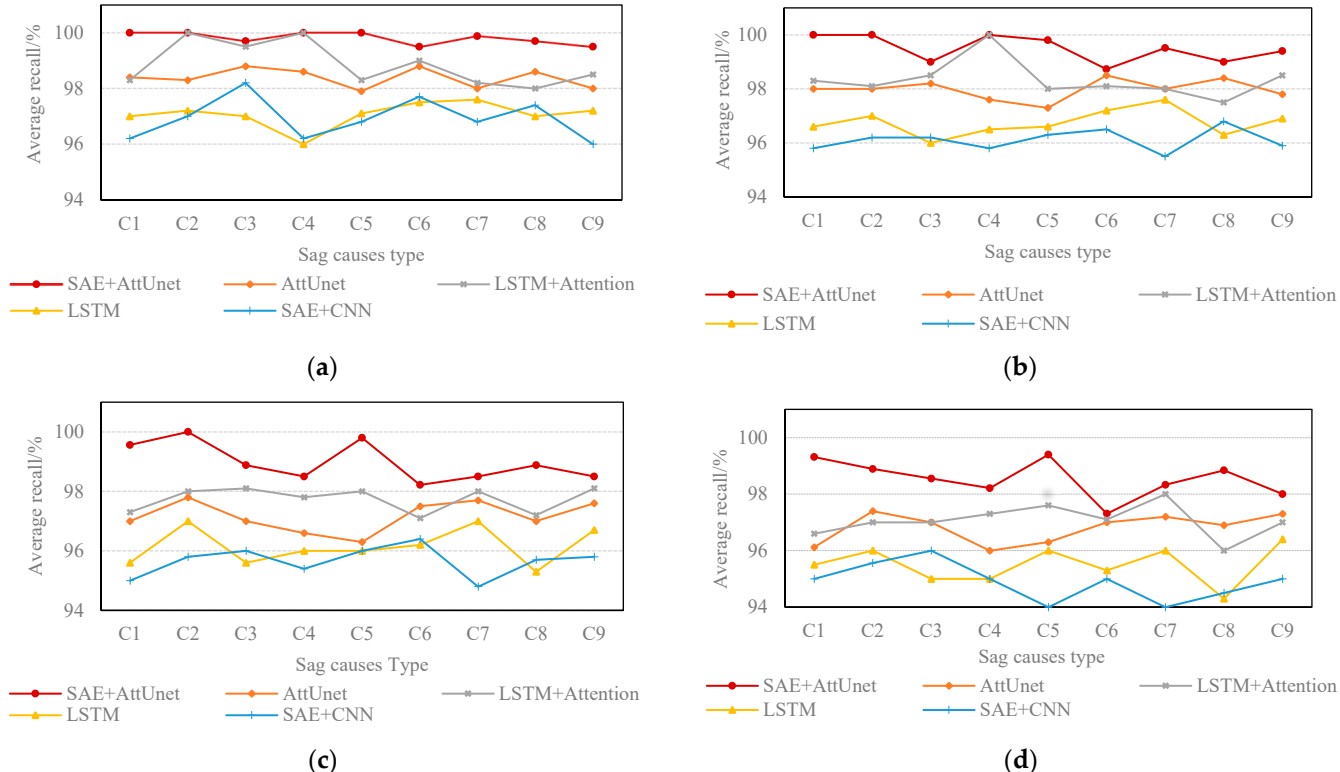

**Figure 10.** Comparison of model average recall under different noises. (**a**) Comparison of recognition accuracy at 50 dB; (**b**) comparison of recognition accuracy at 40 dB; (**c**) comparison of recognition accuracy at 30 dB; (**d**) comparison of recognition accuracy at 20 dB.

The original staging feature is unstable under the influence of noise factors such as power grid conditions, which ultimately affect the judgment ability of the model. The SAE can reduce noise and dimensionality in this model. The encoder maps the original high-dimensional data to the low-dimensional implicit layer vector space to achieve feature dimensionality reduction. The low-dimensional depth feature ensures the high quality of the data at the same time and can also reduce the training complexity of the identification model. AttUnet without SAE cannot distinguish the relationship between the original high-dimensional features very well, and the model training complexity is high, which affects the recognition effect.

This article further describes feature extraction via AttUnet. The AttUnet model is based on the U-shaped network structure. The attention mechanism is added before each hop connection to control the importance of different locations and improve the network sensitivity. At the same time, the layer-by-layer jump connection operation

integrates the attributes of the underlying features and the high-level features, ensuring that the superimposed feature map will not lose important information during the feature extraction process and the distinction between the extracted staging features is further improved. Compared with the CNN model that joins SAE, the AttUnet model has better feature extraction capabilities. The introduction of the attention mechanism can effectively highlight the characteristics that affect the category of the staging signal, and the LSTM of fusion attention has a better recognition effect than the traditional LSTM. However, LSTM cannot distinguish the changing relationship between features very well, thus affecting the final recognition and judgment ability.

## 5. Conclusions

In this paper, a voltage sag recognition method integrating SAE and AttUnet is proposed. The expression of low-dimensional characteristics of hidden variables is obtained by training high-dimensional perturbation data of an unsupervised network SAE, which solves the problems of high computational complexity and weak feature extraction ability of traditional models due to unstable characteristics of the original sample, large volume, and stagnant feature vector dimensions, improves the data processing ability, and significantly reduces the input dimension of the classification recognition model. Then, after combining full-scale feature extraction and supervised training by multiple fully convolutional network AttUnet, the model characteristics of AttUnet are used to synthesize the high-level and low-level feature data in extracting features. The attention mechanism is combined to distinguish between different dimensional elements before jumping and connecting layer by layer so that the fused features are more expressive and distinguished. By identifying nine kinds of staging perturbations, the experiment shows that compared with the traditional recognition methods, the method has higher recognition performance and better noise resistance in different data environments under the same number of iterations, considering the balance of low complexity and high accuracy of high-dimensional sample staging data training.

The voltage sag identification method based on SAE + Unet has specific adaptability and can automatically extract relevant features with environmental changes, reducing manual intervention. The model can adopt incremental training to adapt gradually to the new working environment. With high precision and strong anti-noise, the proposed method can quickly analyze the causes of sag at each monitoring point and provide auxiliary decision-making for power quality management and governance, which is of great significance for solving power quality problems.

**Author Contributions:** Investigation, R.F.; Data curation R.F., H.L. and T.Z.; Study design, R.F. and H.L.; Software, T.Z.; Writing—original draft preparation, R.F., H.L., T.Z. and H.W.; Writing—review and editing, L.Q. and L.S., Figure, T.Z. and L.S. All authors have read and agreed to the published version of the manuscript.

**Funding:** This work was supported by State Grid Shanxi Electric Power Company Science and Technology Project Research 520530200011.

**Data Availability Statement:** Most of the data are not applicable; simulated data can be provided upon request.

**Acknowledgments:** The authors greatly appreciate the reviews, valuable suggestions, and editors' encouragement.

**Conflicts of Interest:** The authors declare no conflict of interest.

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
