# Peer review of "Voltage Sag Causes Recognition with Fusion of Sparse Auto-Encoder and Attention Unet"

_electronics, doi:10.3390/electronics11193057_

Round 1

Reviewer 1 Report

This paper presents a combination of sparse auto-encoder and Attention Unet for voltage sag causes recognition. However, the paper should be improved before it can be accepted by considering the following comments:

1) The title is misleading content of the paper when the word "causes" is missing. I believe the paper is about identifying the causes of voltage sag and not identifying voltage sag from bad power quality voltage signals.

2)  Labels a, b, c (stages) in Figure 1 are not explained and why they are there? Different lines are presented in the figure (dashes, solid and dots) and they are not explained as well.

3) It is unclear how the recognition of voltage sag causes is considered as a high-dimensional feature problem.

4) Formatting issues should corrected such as variables in text are not italic, font size of Eq (4), capital letters in the middle of sentences, etc.

5) Notation i is used in many places and different things. It represents a group of samples in Eq (3), neuron in Eq (4) type of recognition in Eq (6) and sample in Eq (7). It is good to use other alphabet to differentiate them.

6)  Short-circuit fault simulation data collection is unclear - First sentence of first paragraph starting at line 292 in page 8 is unclear. Where is the location of the fault occurrence in IEEE 14 node and where the measurement takes place?

7) How do you get 500 groups from the data? What are the different between them?

8) It is good to present a sample of the simulated short circuit fault data (one for each cause)

9) Please explain why the accuracy in Table 1 is improved after noise is set to 30 dB and above. Why does it only drop at 20 dB?

10) Label and unit for y-axis in Figure 8 are missing. Figure 8(a) contains 2 blue dots, what are they? What is the purpose of showing results at different noise level in the figure but they are not explained in text?

11) The paper doesn't present how much improvement of accuracy in numerical values that can be achieved using proposed method as compared to using SAE or AttUnet alone.

12) Conclusion should relate to the important of identifying voltage sag causes from the obtained results.

Reviewer 2 Report

The paper introduces another approach to voltage sag identification based on a combination of a sparse autoencoder and "attention unet", the latter providing the novel point of the work compared to other approaches.

The major difficulty with this topic is to compare consistently with previous literature and published results, providing a fair comparison of performance. This is the major point on which the following comments focus.

1) Line 27. You mention a "large number" of events and "severe impact", but then there is only one reference cited for this. In the last comment on References you can see two suggestions for review papers R1 and R2. Please, include some relevant references.

2) Line 33. "The high-dimensional sample voltage sag data" seems referred to ref [3]. You should characterize better this data set, in terms of how many features, how many recordings, quality of recordings (such as if full waveforms were recorded and stored, or just for example peak values and duration, etc.)
In addition, are these data accessible?

3) Line 39. What is the meaning of "at home and abroad"?

4) Line 40 and following.
a) You should improve the clarity of the classification you propose which divides approaches into two broad classes, because this is part of the demonstration that you comprehensively consider all previous relevant results and approaches.
b) You should increase the extent of the overview and the depth of your review, possibly including for example the two reviews R1 and R2.
c) In addition, it is advisable that you provide a broader view of past approaches, including most traditional methods to voltage identification, such as including the extracted references D1 to D5. Both "R" and "D" references are clarified in the last comment.

5) Figure 1. It is not a comparison "chart", and it is very small that cannot be read. It seems also taken from some other publication, so -- if it is the case -- you should indicate your source and taking care of copyright issues.

6) Section 3 title. You may remove "Concrete" that is redundant.

7) Figure 5. To improve legibility, including a uniform size of fonts.

8) Section 4. You should report some of the voltage sag waveforms and how the added noise look like.

9) Section 4.1. The voltage sag waveforms you use are simulated and not real measurement results. Please, clarify how such simulated waveforms are really representative of a real scenario, where you might have deformation of waveforms, phase jumps, and so on (besides harmonic distortion that is commented elsewhere).

10) Line 309 and 366. You speak of noise and signal-to-noise ratio. In the task of detecting and recognizing voltage sags harmonic components are particularly important, rather than incoherent white noise (like yours, Gaussian distributed), that is not so relevant.
In some of the suggested references there are explicitly examples with significant amount of harmonic distortion, that usually should be considered in the order of max 5% for distribution networks and 10% for industrial networks, of course intending maximum voltage distortion, and not current, that may be much higher.

11) Eq (8) is said to be accuracy, but uses the size of the sets and not an approach based on the classification results and thus based on a confusion matrix. I don't think it is correct. Could you provide explanations and clarifications in the paper?

12) Eq (9) is again "accuracy" like equation (8). Here you speak of "correct" sample data, that is not clear. Does it mean correctly classified? What is the difference with respect to eq (8)? You should use different symbols for (8) and (9) and also for eq (9) you should clarify exactly the terms of the problem, that is what is exactly S and N. There is a commonly accepted approach based on confusion matrix and derived indexes (e.g. balanced accuracy) for classification problems.

13) Figure 8. It seems that you verify that voltage sags are correctly labelled, but you do not use waveforms that has shallow sags (that are not sags, being above the 90% threshold) to verify that they are not misclassified. The same is for waveforms with significant distortion, that might be misclassified (harmonic distortion changes the instantaneous value when the voltage sag begins and thus may cross the 90% threshold although the sag is above 90%).

14) References are incomplete and focused on a few Chinese Journals rather than including the large knowledge repository on the subject of voltage sags that is IEEE (in particular Trans. on Power Delivery) and MDPI itself (in particular Energies).
In the following I exemplify references that are suitable for the Introduction, to improve the presentation of the state of the art and to highlight novelty and performance of the proposed approach.

Reviews:
R1] A.Khoshkbar Sadigh ; K.M.Smedley "Fast and precise voltage sag detection method for dynamic voltage restorer (DVR) application", Electric Power Systems Research, 2016. https://doi.org/10.1016/j.epsr.2015.08.002
R2] G. W. Chang;Cheng-I Chen "Performance evaluation of voltage sag detection methods", IEEE PES General Meeting, 2010. 10.1109/PES.2010.5589426

Other prominent methods with comparable (possibly better?) performance, including accounting for distorted waveforms:
D1] C. Fitzer;M. Barnes;P. Green, "Voltage sag detection technique for a dynamic voltage restorer", IEEE Transactions on Industry Applications, 2004. 10.1109/TIA.2003.821801
D2] A. Florio;A. Mariscotti;M. Mazzucchelli, "Voltage sag detection based on rectified voltage processing", IEEE Transactions on Power Delivery, 2004. 10.1109/TPWRD.2004.829924
D3] Y. Kumsuwan;Y. Sillapawicharn, "A fast synchronously rotating reference frame-based voltage sag detection under practical grid voltages for voltage sag compensation systems", 6th IET International Conference on Power Electronics, Machines and Drives (PEMD 2012). 10.1049/cp.2012.0348
D4] Yutthachai Sillapawicharn;Yuttana Kumsuwan, "Dual low pass filter-based voltage sag detection for voltage sag compensator under distorted grid voltages", 2014 International Electrical Engineering Congress (iEECON). 10.1109/iEECON.2014.6925913
D5] Li, Z.; Yang, R.; Guo, X.; Wang, Z.; Chen, G. A Novel Voltage Sag Detection Method Based on a Selective Harmonic Extraction Algorithm for Nonideal Grid Conditions. Energies 2022, 15, 5560. https://doi.org/10.3390/en15155560

Round 2

Reviewer 2 Report

Dear Authors, thank you for your replies.

I do not have other comments.